# Changing the Paradigm-Controlling Polymer Morphology during 3D Printing Defines Properties

**DOI:** 10.3390/polym14091638

**Published:** 2022-04-19

**Authors:** Daniel P. da Silva, João Pinheiro, Saba Abdulghani, Christina Kamma Lorger, Juan Carlos Martinez, Eduardo Solano, Artur Mateus, Paula Pascoal-Faria, Geoffrey R. Mitchell

**Affiliations:** 1Centre for Rapid and Sustainable Product Development, Polytechnic of Leiria, 2430-080 Marinha Grande, Portugal; daniel.p.silva@ipleiria.pt (D.P.d.S.); joao.d.pinheiro@ipleiria.pt (J.P.); saba.abdulghani@nms.unl.pt (S.A.); artur.mateus@ipleiria.pt (A.M.); paula.faria@ipleiria.pt (P.P.-F.); 2NCD-SWEET Beamline, Alba Synchrotron Light Source, Cerdanyola del Vallès, 08290 Barcelona, Spain; ckamma@ansto.gov.au (C.K.L.); guilmar@cells.es (J.C.M.); esolano@cells.es (E.S.)

**Keywords:** 3D printing, polymer texture printing, polymer morphology, crystal orientation, mechanical properties

## Abstract

Direct digital manufacturing consists of a set of techniques that enable products to be fabricated directly from their digital definition, without the use of complex tooling or moulds. This manufacturing approach streamlines prototyping and small-scale production, as well as the mass customization of parts with complex designs immediately fixed before fabrication. With broad applicability, there are clearly opportunities in the field of medical devices for its use. However, many of the developments of direct digital manufacturing focus on simply specifying the shape or the form of the product, and this limited scope throws away many of the particular advantages of direct digital manufacturing. This work is focused on remedying this situation so that the digital specification of the fabricated product includes the properties as well as the form of the product. We use in situ time-resolving small-angle X-ray scattering measurements performed at the ALBA Synchrotron Light Source in Barcelona to evaluate the control that can be exerted on the morphology of a semi-crystalline polymer during extruder-based 3D printing. We use this as a methodology for printing the patterns of the morphology of the polymer to realise the patterns of properties of the polymeric material, specifically the modulus of the polymer. We give an example of products produced in this manner that contain spatial variation in their properties.

## 1. Introduction

Direct digital manufacturing is a family of technologies that enable products to be produced from a digital definition directly, without the use of specialised tooling or moulds [1]. Such technologies include stereolithography and fused deposition modelling; the latter is the focus of this work. These technologies have developed out of methodologies developed specifically for rapid prototyping in which the external shape or form of the product was the focus.

Although the realisation that products could be produced from materials with suitable properties for application has been widely adopted in additive manufacturing, there has no real effort to further develop the technology to include patterns or the texture of properties so as to be able to deliver the function of the product at the point of fabrication. Of course, there are other advantages of additive manufacturing, and the abilities to control the volume of material used by changing the infill pattern and to optimise the design of the product through topological optimisation [2] have played a significant role in the design of products using FDM.

## 2. Three-Dimensional Printing

Fused deposition modelling, commonly referred to as 3D printing, was developed by S. Scott Crump, co-founder of Stratasys, in 1988 [3]. In this process (Figure 1), a thermoplastic strand is heated to the liquid state and fed through a small extruder and emerges as a viscous liquid strand to be deposited on a moving build platform, where it cools to retains its shape on solidification through vitrification or crystallisation. The print head moves relative to the build platform in order to deposit material where it is required. The three-dimensional structure is built up layer by layer. Since that time, variants have emerged that are based on pellet-fed extruders [4] and those that process a fluid suspension to print ceramic and cement-based products [5], as well as hybrid systems, which print a metal powder in a polymer matrix, wherein the matrix is burnt away, leaving a metallic part [6].

Whatever the design, for plastics, a fluid polymer is passed through a nozzle, starting with a large diameter and then a small diameter nozzle with a particular length-to-diameter ratio. Although it is most likely that the polymer jet swells as it emerges from the nozzle [8], the nozzle diameter in a large part defines the strand diameter that is deposited on the build platform, and this, in part, defines the resolution of the print process. Now, it is well known that passing a fluid polymer through a restricted die will lead to flow effects involving shear and elongational flow [9].

For a structured material such as a liquid crystal polymer, in which domains with different director orientations exist in the quiescent fluid, passing through the die leads to a common alignment of these director patterns. This is shown schematically in Figure 2, and Figure 3 shows the wide/angle X-ray scattering pattern for an extruded pellet of a thermotropic liquid crystal polymer [10].

An intense equatorial peak at Q ~ 1.3Å^−1^ can be observed. This arises from interchain correlations, and the limited spread of this peak indicates a high level of preferred orientation in the extruded pellet. The fact that this interchain peak is centred on the equatorial section indicates a high level of preferred orientation of the polymer chains parallel to the extrusion axis. The extent of the arcing arises from the level of orientation, which is a convolution of the nematic order parameter within a domain and the orientation distribution of the director of each domain. In the example shown here, we expect that the pellet is essentially a nematic mono-domain and, hence, the spread is related to the order parameter.

This high level of preferred molecular alignment will have a large impact on the physical properties of the resultant solid part. The part will be much stiffer in the direction parallel to the extrusion direction than an isotropic structure.

For an amorphous polymer, the level of preferred orientation induced by flow will be much less marked, as it depends on chain deformations between entanglement points and will strongly depend on the molecular weight distribution of the polymer. For a semi-crystalline polymer, the situation is rather different.

The level of preferred orientation in the melt phase during shearing has been shown to be very low at 〈P_2_〉 ~ 0.01 using in situ wide-angle X-ray scattering measurements [12] and via small-angle neutron scattering on isotopically labelled mixtures [13].

Essentially, the longest chains support the stress and become elongated in flow, and the remainder of the melt is in a relaxed state. On reaching a temperature at which crystallisation occurs, if these chains are still present in the elongated state, they act as row nuclei for the remaining polymer melt to crystallise, and chain-folded lamellar crystals grow out from the row nuclei with the growth direction normal to the elongated chains and, therefore, the flow direction.

The massive impact of this templating was shown dramatically by contrasting the morphology of a sample of a branched polyethylene (*M*_w_ = 76,800) crystallised from a sheared melt, which exhibited no preferred orientation, with that of a blend of the branched polyethylene with 10% linear polyethylene (*M*_w_ = 312,000) [14]. The resultant microstructure shows a massively high level of preferred orientation through the sample examined. Clearly, the two morphologies shown in Figure 4 will exhibit very different physical properties. We anticipate observing very similar effects as a consequence of 3D printing using different process parameters.

Others have made observations of the development of preferred orientation during extrusion [15], although not using in situ techniques, and [16], whose work was restricted to WAXS studies, provides ambiguous information on the preferred orientation of the lamellar crystals. If we can adjust the process parameters in real time during printing, then we will have a method for depositing patterns of different morphologies or different properties. This is the focus of this work.

## 3. Materials and Methods

The current work is focused on poly(e-caprolactone) (PCL), a widely studied biodegradable material with biomedical applications, including as scaffolds for tissue engineering [17]. It was supplied in the form of small pellets (~3 mm) with an *M*_w_ = 50,000 by Perstorp (Cheshire, UK).

The thermal characteristics of the PCL were determined using a Perkin Elmer STA 6000. All operations were performed under a nitrogen atmosphere with a flow rate of 20 mL/min, and the weight of the sample varied between 6 and 7 mg. Each sample was heated first to 120 °C at 10 °C/min and then held at that temperature for 1 min to erase any thermal history. The crystallisation temperature was obtained from an equivalent cooling scan at 5 °C/min.

Small-angle X-ray scattering (SAXS) and wide-angle X-ray scattering (WAXS) measurements were performed at the ALBA Synchrotron Light Source in Barcelona using the NCD-SWEET Beamline [18]. The beamline provides facilities for measuring both SAXS and WAXS simultaneously. SAXS patterns have been taken in a q-range from 0.017 Ᾰ-1 to 0.125 Ᾰ-1 and the WAXS ones from 1.0 Ᾰ-1 to 3 Ᾰ-1. The SAXS detector was a Pilatus3S 1M system from DECTRIS, which is a hybrid single-photon counting system. The absorption of the X-ray photon in the detector leads to the formation of electron–hole pairs and a charge which is proportional to the photon energy.

The Pilatus system is built up of an array of silicon sensors equipped with CMOS electronics. As a consequence, a small portion of the detector is not active (~7%), which appear as black stripes in the recorded intensity images. The charge is detected and processed by the pixel readout system. This detector has an effective pixel size of 172 × 172 µm and a dynamic range of 20 bits. To prevent the saturation of the detector by the zero-angle X-ray beam, a beam stop is placed in front of the detector to absorb the transmitted beam. The sample-to-SAXS detector distance was 6.73 m with an incident X-ray wavelength of 1 Ᾰ. The detector orientation and sample-to-detector distance was calibrated using the well-known standard silver behenate. The WAXS detector was a Rayonix LX255 HS, which is a triple-cooled CCD detector bonded via fibre optic tapers to the X-ray photon detector surface with a pixel size of 44.27 × 44.27 µm. The detector geometry and shape ensure that the direct beam and the SAXS pattern are not blocked by the detector. The WAXS orientation and sample-to-detector distance was calibrated using Cr_2_O_3_. For each 2d SAXS pattern, an azimuthal section *I*(α) was obtained at constant |Q| and as a function of α, where α is the angle between the extrusion axis, which was vertical to the beamline, and the scattering vector Q. The azimuthal section *I*(α) was used to evaluate the level of preferred orientation of the chain-folded lamellar crystals using the methodology developed by Mitchell [19,20,21], which is shown below.
(1)$${\langle P_{2n}\rangle }_{Q}=\frac{1}{\left(4n+1\right)P_{2}^{\;m}}\overset{\pi /2}{\underset{0}{\int }}\frac{I\left(\left|\underline{Q}\right|,\alpha \right)\sin\;\alpha P_{2}\;(\cos\alpha )\;d\alpha }{I\left(\left|\underline{Q}\right|,\alpha \right)\sin\alpha \;d\alpha }$$

The orientation parameter 〈P_2_〉 is the first component of an even series that describes the orientation distribution function of the normal crystals to the lamellar crystals. If 〈P_2_〉 = 0, then the distribution is isotropic; if 〈P_2_〉 = 1, then the crystals are arranged with the same perfect preferred orientation

## 4. Results and Discussion

### 4.1. Thermal Analysis 

The thermal analysis of the quiescent material was performed in preparation for the in situ experiments, and Figure 5 shows the DSC cooling scan for the poly(ε-caprolactone used in this work. We can observe that in the quiescent material, the onset of crystallisation occurs at 37.9 °C, while the maximum rate of crystallisation is observed to be at a temperature of 35.5 °C. We need to note that the material that has been extruded may start to crystallise at a higher temperature than observed with the quiescent material due to the presence of flow-induced row nuclei.

### 4.2. Three-Dimensional Printer

The 3D printer was mounted on the NCD-SWEET beamline using the standard user platform, which could be adjusted in all three directions. The 3D printer was specifically designed for these experiments, and a schematic is shown in Figure 6.

The system shown in Figure 6 enables the extruder to be operated with a constant extrusion temperature, a varied extrusion velocity, and a variable write velocity. The latter is the relative motion of the build platform to the extrusion nozzle.

The equipment uses a single extruder with a dual-channel screw to push the material through a 300 μm needle. Then, the material is deposited onto a rotating collector. In this case, the extruder was designed to intake material in the form of beads or solid grains, instead of filament, which is more commonly used in fused deposition modelling (FDM); thus, the material reservoir is heated to melt the polymer material in order to feed the extruder with molten polymer. This configuration requires temperature monitoring of the three main sections of the extruder: the reservoir, the feeder connection, and the extrusion chamber. In each of these zones, we assembled an independently controlled coil heater with an integrated thermocouple sensor and feedback loop. This is a similar design to that used in the bioextruder [4], which was also developed at CDRSP. The option of defining the three temperatures allows for precise control of the temperature gradient of the material throughout the extrusion process. Appendix A provide details of the 3D printer and its specification.

In order to meet the requirements imposed by the NCD-SWEET beamline, we have designed a specific printer for this purpose (Figure 7), particularly taking account of:The boundaries of the mounting platform;The operability at the beamline;The process stability;The extruder and collector positioning;The range of extrusion rate;The thermal insulation.

As the boundaries of the sample stage are limited (200 mm × 200 mm), the equipment, consisting of the extruder, the collector, and the supporting structure, constitutes a compact solution that can be adjusted in all three directions. The collector is a small winding mechanism powered by a stepper motor. We have opted for a rotational build platform rather than the normal translation stage, as it offers a more compact solution and, also, it enables higher write speeds, if these are required. The extrudate is deposited on the surface of a roll that simulates the continuous translation of an extruder during FDM, maintaining the melted polymer flow in the same direction and the stable positioning of the filament for data acquisition.

The limited space for the experiment is a major condition since, ideally, the detector and the vacuum chamber should be as close as possible, to reduce the contribution of the air scattering in the sample region to the background.

Simultaneously, it is necessary to provide enough space for a safe operation, which includes not only the disposition of the apparatus elements, but also the freedom to perform the tasks of the experiment: feeding the system with raw material, cleaning and removing deposited material, and replacing extruder dies, among other maintenance actions. This is a particular concern when working with high temperatures near the beamline equipment, and thus the heated zones of the extruder in contact with the supporting structure are insulated with a 6 mm heat insulation board.

Crucial input instruments, such as the power supply, the heater-controlling modules, and the pneumatic actuation of the extruders’ feeder (as shown in Figure 8), were placed on a table adjacent to the beamline.

During the analysis, direct temperature measurements were performed using an infrared (IR) thermal camera at a distance of 200 mm from the sample. Processing variables, such as screw rotation velocity, write velocity, processing temperature, and needle-to-collector distance, were fully remotely controllable, with values in the range of the normal operation of 3D printers. At the beamline, the stability of the position of the extrudate during extrusion is mandatory from the first instance of each trial. In this specific experiment, the sample was only available while the extruded material was flowing, and vibrations or irregular portions of the material deposited on the collector negatively interfered with this stability and limited the duration and quality of the test. Long trial durations were required to perform a vertical scan along the height of the extruded filament, which involved moving the sample platform in the z-axis, maintaining the filament aligned with the X-ray beam, by using the collector presented in Figure 9.

The smooth movement of the filament was induced by the stable rotation of the main roll, made of a 50 mm diameter acrylic tube, with input power from a 3:1 timing belt transmission from the motor’s axis. Once at low rotations, the vibrations induced by the stepper motor are significant and, therefore, the angular velocity of the motor was increased to reduce the impact of vibrations, allowing the test to proceed with a range of write speeds from 5 to 50 mm/s.

The remaining two rolls were also important for the sample quality. Powered by the smallest roll (in the driving shaft, also working as a gear), the middle roll has a rough surface from which to remove the filament from the main roll via contact, and a gap to pressure it and secure it during the winding, which ends at the third roll. These components were manufactured in ABS using FDM printing technology. Structural and functional parts were rapidly produced to generate a modular configuration with custom parts in which the position of the extruder and the position of the collector could be adjusted. For instance, the distance between the needle and the collector is variable between 0 mm and 50 mm, meaning that the X-ray beam can traverse the length of the sample filament. With this degree of freedom, experiments were performed by continuously extruding a steady-state polymer filament and acquiring simultaneous SAXS and WAXS patterns at varying distances from the needle.

### 4.3. Three-Dimensional Printer Evaluation

To explore these concepts, we have developed the pellet feed extruder-based 3D printer shown in Figure 6 (left) which could be mounted on the NCD-Sweet beamline at the ALBA Synchrotron Light Source in Barcelona in order to observe the structure and morphology in real time using small-angle and wide-angle X-ray scattering techniques [22].

Figure 10 shows a plot of the temperature of the extrudate as a function of the distance from the outlet of the extruder die. The plotted values correspond to an extrusion rate of 300 mm/min. The graph shows that it takes much longer for the extrudate to cool from an extrusion temperature of 75 °C to the temperature at which the maximum crystallization rate occurs than for an extrusion temperature of 45 °C or 37 °C.

The longer cooling time is sufficient for the extended chains induced in the extruder die to relax to an isotropic random coil configuration. The shorter cooling times ensure that the extended chains have not relaxed at the point at which crystallization is initiated. These data are specific to PCL, and other polymers will have differing extrusion temperatures and temperatures at which the maximum crystallization rate is observed.

The small-angle X-ray scattering provides information about the chain-folded lamellar crystals and the level of preferred orientation with respect to the extrusion axis. If we position the X-ray beam close to the extrusion nozzle, then, as can be seen in Figure 11 and Figure 12(right), the SAXS pattern is featureless, as the sample is molten and contains no electron density variations on this scale. If we move the X-ray beam further down the extrudate, the temperature has started to drop, and when the crystallization temperature is reached, crystals start to grow.

Figure 11 and Figure 12 show patterns taken at different points along the extrudate. The only difference between the two sets is the write speed, which is lower in Figure 11 and higher in Figure 12. The right-hand pattern in Figure 11 shows that the solidified extrudate has a more or less isotropic crystal orientation, whereas the equivalent pattern in Figure 12 shows a high level of orientation, as revealed by the two sharp maxima above and below the zero-angle point with a small level of isotropic scattering.

Now, in the extruder nozzle, the longest chains may be extended in the flow field, while the remaining polymer is unstressed. If the extrudate reaches the crystallization temperature and these extended chains are still present, they serve as row nuclei and templates for the crystallization of chain-folded lamellar crystals, which grow out normal to the extrusion direction [23]. If, on the other hand, the extended chains relax and adopt a random conformation, then crystals are nucleated by other heterogeneities and an isotropic spherulitic morphology results. These two possibilities are shown schematically in Figure 13. Previous studies [22] showed that the moduli of these two different morphologies differed by more than a factor of 2.

This difference arises directly from the change in writing speed. The very high level of orientation of the lamellar crystals shown in Figure 12 is very typical of row-nucleated crystallization, as the row nuclei provide a common, highly aligned templating mechanism. It might be thought that these effects are caused by the deformation of the printed strand by the relative movement of the build platform with respect to the extruder die. In such a scenario, we would see the crystallization of lamellar crystals with a low level of orientation, which would increase in time as the filament is deformed.

There is no evidence for this mode of behaviour, as is clear from Figure 12, which shows a high level of orientation of the first formed crystals. We attribute this behaviour to the fact that the higher write speed leads to faster cooling as the printed material is moved away from the hot region close to the extruder more rapidly, meaning that crystallization takes place before the extended chains have relaxed.

Switching between these two different morphologies could be achieved by changing the extrusion temperature, although, of course, this requires a certain amount of time to equilibrate. We also found that the overall level of preferred crystal orientation in the deposited material could be changed continuously using a combination of the extrusion speed and the write speed. The latter is the relative motion of the build platform to the extrusion nozzle. 

This behaviour is confirmed in Figure 14. Quantitative values can be evaluated directly from the recorded small-angle X-ray scattering patterns in a fully rigorous manner [22]. The vertical scale is the orientation parameter <P_2_>, which can take values from 0.0 (corresponding to an isotropic morphology) to 1.0 (corresponding to a uniform orientation of crystals).

The plot of orientation parameters derived from the patterns in Figure 14 (triangles) shows that the initial orientation is high, but as the level of crystallinity increases, the average orientation reduces in value. Of course, the highly aligned crystals that were first formed are still present in the final product. In contrast, the level of orientation of the crystals formed during the lower write speed (Figure 11 and Figure 14 (squares)) is lower, but increases as more crystals form. In these plots, we underline the value of in situ measurements for revealing the complex crystallisation processes taking place during printing.

As an example of the use of this new approach to modulating the properties by controlling the morphology, we show in Figure 15 a photograph of a hexagonal open structure typical of that used as a scaffold for tissue engineering.

The part is fabricated from PCL, as used in the previous experiments, and prepared using a pellet-fed extruder-based printer, in which the extrusion rate and temperature were fixed throughout the fabrication, but the relative speed between the print head and the build platform was varied as a function of the radial distance between the centre of the object and the print position.

This can be easily achieved through a suitable modification of the g-code [24] file used as an input to the printer. As a consequence, the central part of the product is softer than the outer part due to the differing morphologies of the printed PCL.

One limitation of this approach is that the anisotropy, in the study performed here, only develops in the direction of the extrudate axis and, therefore, follows the pathway of the print head. This needs to kept in mind when designing a product and the tool path. We are currently exploring such aspects, and we have observed some interesting phenomena, and we will present the outcomes of such studies when completed in a future paper.

## 5. Conclusions

We have shown that the process of extrusion inherent in fused deposition modelling technology can be controlled using the natural processing parameters to produce deposited polymeric material with different morphologies that exhibit different physical properties.

As some of these process parameters can be varied in real time during the printing process, essentially on the fly, this provides a method of depositing the polymer with different properties in different zones of the product.

Exploiting time-resolved X-ray scattering studies of in situ 3D printing, we have mapped out the correlations between the print parameters and the level of preferred orientation in the deposited material.

We have used this understanding to prepare a part with a design similar to that of a porous scaffold for use in tissue engineering, with a gradient of properties of the printed part highlighted in a radial manner. The variation and control of properties in this example exists in the direction parallel to the tool path.

We are currently exploring the design of products to take advantage of this new approach. Preliminary studies have revealed some interesting results, and these will be presented in a future paper when the work is completed.

## Figures and Tables

**Figure 1 polymers-14-01638-f001:**
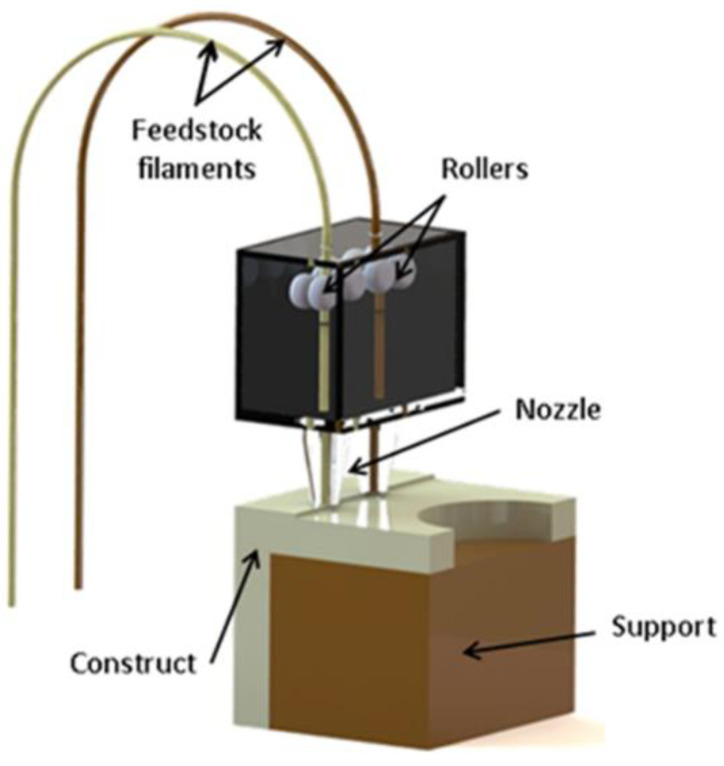
A schematic representation of the essential elements of fused deposition modelling technology. Reproduced with permission from [7].

**Figure 2 polymers-14-01638-f002:**
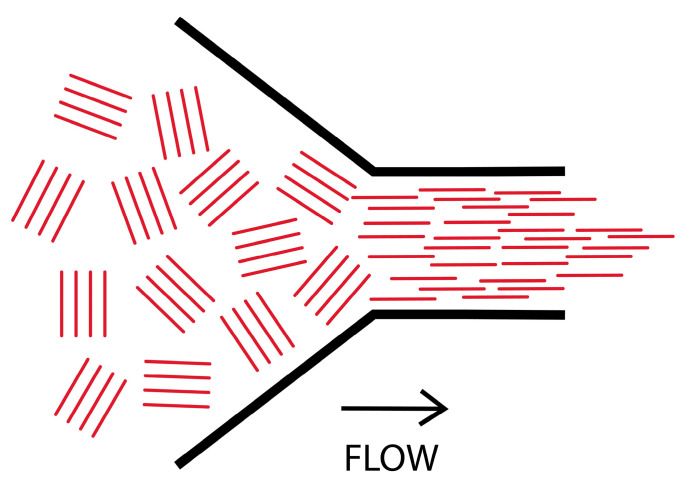
A schematic of liquid crystal fluid passing through a restricting die. The lines represent the local liquid crystal director. Reproduced and adapted from [11].

**Figure 3 polymers-14-01638-f003:**
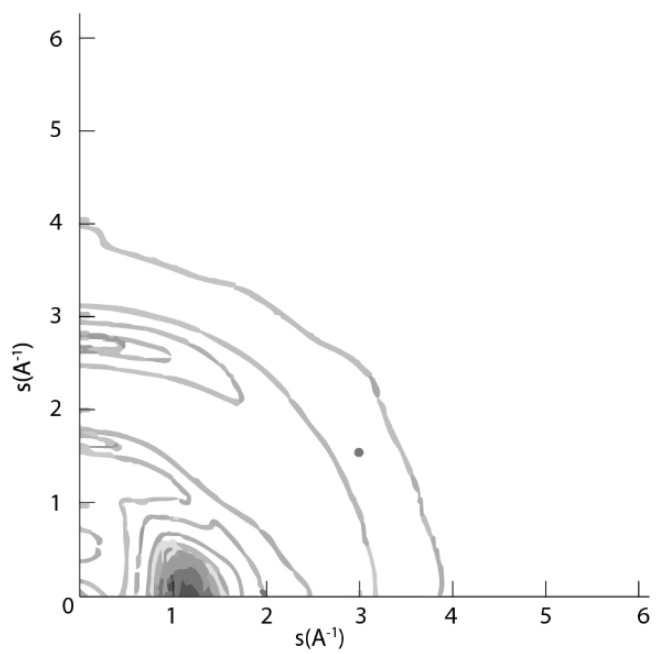
The wide-angle X-ray scattering pattern recorded for an extruded pellet of a thermotropic liquid crystal polymer. The extrusion axis is vertical (s = 4πsinθ/λ). Reproduced with permission from [10].

**Figure 4 polymers-14-01638-f004:**
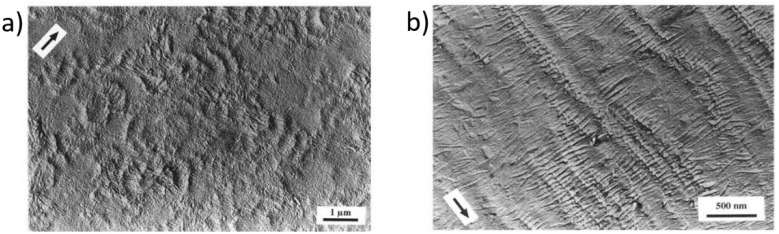
Transmission electron microscopy images of replicas of differentially etched samples of (**a**) branched polyethylene subject to shear flow at 25s^−1^, wherein the arrow shows the shear direction, and (**b**) a blend of the same branched polyethylene containing 10% linear polyethylene crystallised at 115 °C, also subjected to shear at 25s^−1^. Reproduced with permission from [13].

**Figure 5 polymers-14-01638-f005:**
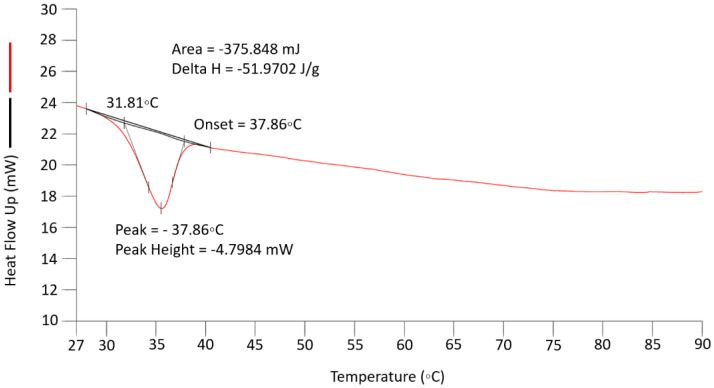
The cooling scan of a sample of PCL used in this work from 120 °C to room temperature showing the crystallisation peak, which reveals that the maximum crystallisation rate for a quiescent melt is 35.5 °C.

**Figure 6 polymers-14-01638-f006:**
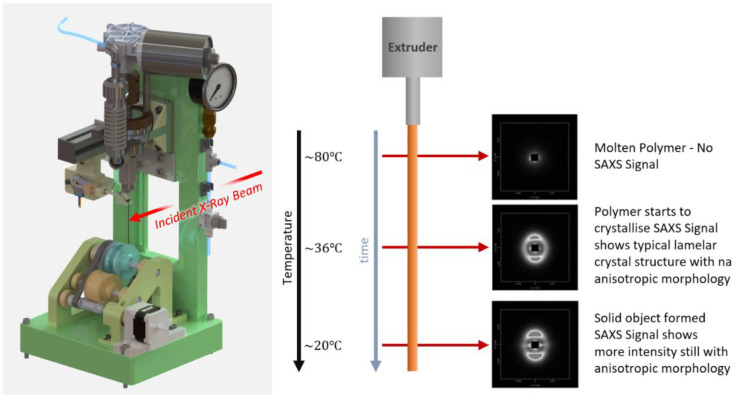
**Left**: a CAD representation of the 3D printer developed for this work. **Right**: a schematic of the quasi-static state of the extrudate in a constant gradient of temperature. The evolution of the structure can be evaluated by moving the incident X-ray beam down the jet.

**Figure 7 polymers-14-01638-f007:**
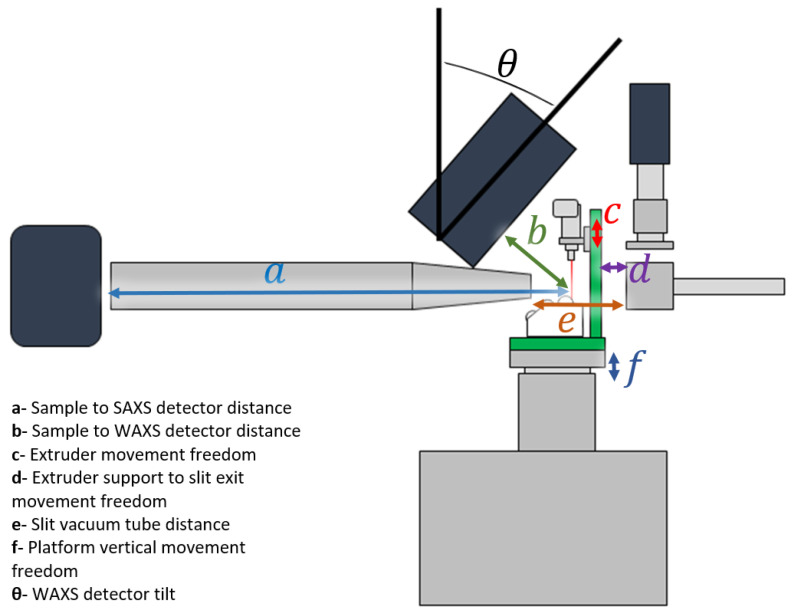
Schematic of the 3D printer system mounted on the NCD-SWEET beamline at the ALBA Synchrotron Light Source.

**Figure 8 polymers-14-01638-f008:**
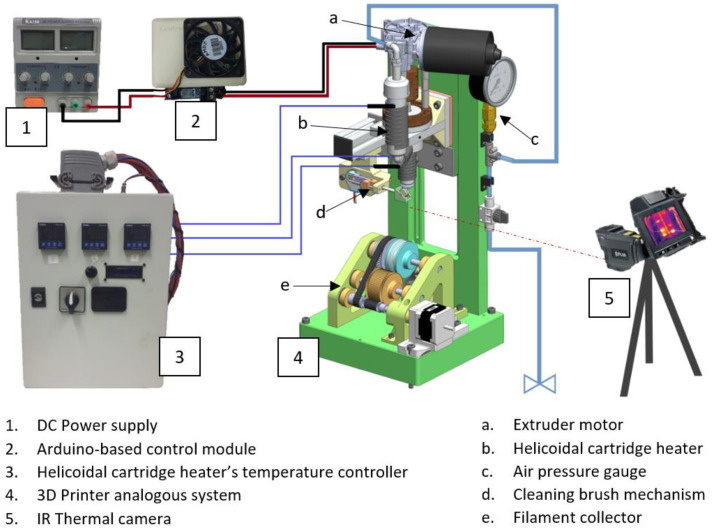
Schematic of the system with the control equipment for the experimental assembly.

**Figure 9 polymers-14-01638-f009:**
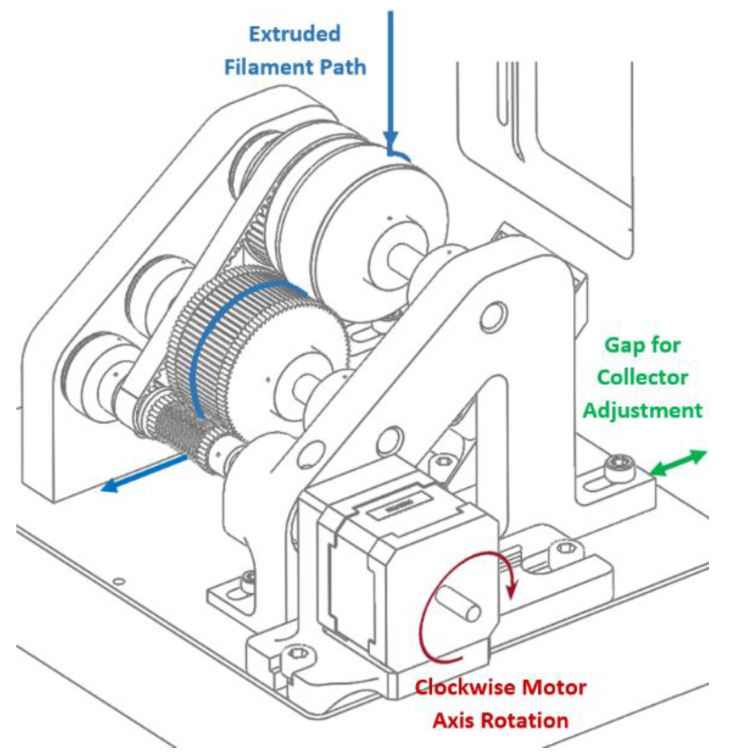
Configuration of the filament collector/build platform.

**Figure 10 polymers-14-01638-f010:**
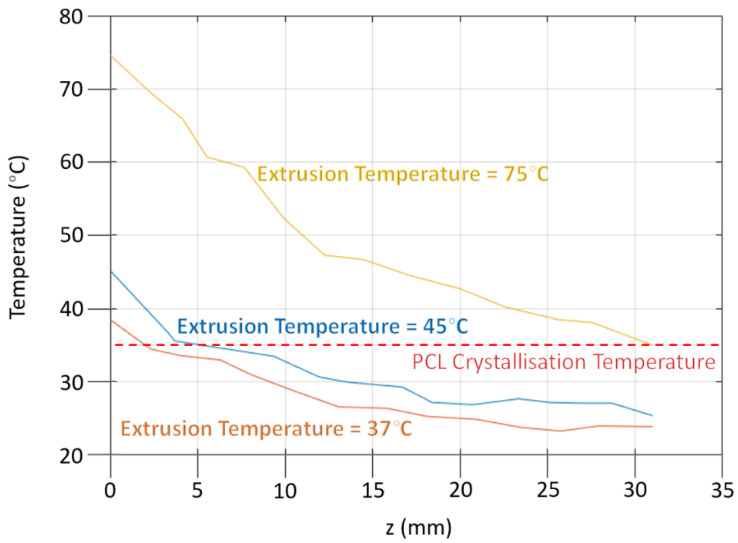
Plots of the temperature of the extrudate after it emerges from the extruder die, measured for different temperatures of the extruder using an infrared camera.

**Figure 11 polymers-14-01638-f011:**
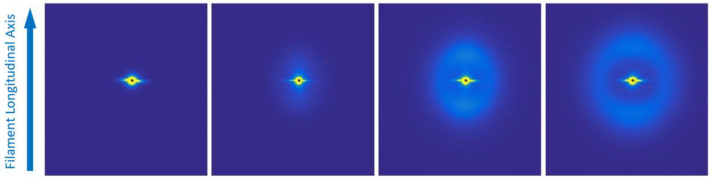
SAXS patterns of the PCL extrudate measured with the incident X-ray beam positioned at increasing distances from the exit of the extruder die (x = 0 mm). The left-hand image is closest to the extruder die, and those to the right are positioned further away. The vertical arrow defines the extrusion axis. The black spot in the centre of each image corresponds to the area of the detector masked by the absorbing beam stop to prevent the forward beam flooding the detector. For all of these images, the 3D printer parameters were fixed for extrusion temperature, extrusion speed, and ambient temperature, and with a low write speed.

**Figure 12 polymers-14-01638-f012:**
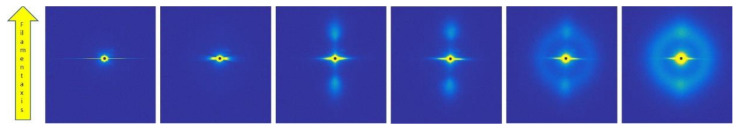
An equivalent set of SAXS patterns of the PCL extrudate measured with the incident X-ray beam positioned at different distances from the exit of the extruder die (x = 0 mm) and prepared with the same parameters as those shown in Figure 11, but using a higher write speed.

**Figure 13 polymers-14-01638-f013:**
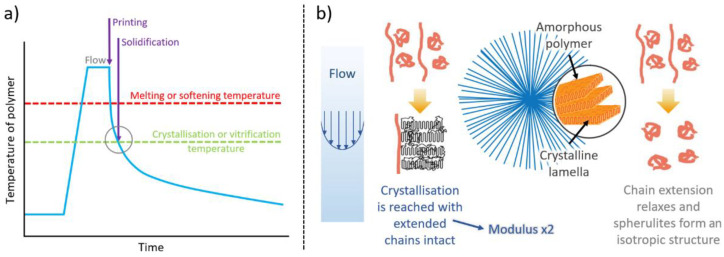
(**a**) A schematic of the temperature time curves relevant to 3D printing; (**b**) a schematic of the process of developing different morphologies with different properties by controlling the 3D printing parameters.

**Figure 14 polymers-14-01638-f014:**
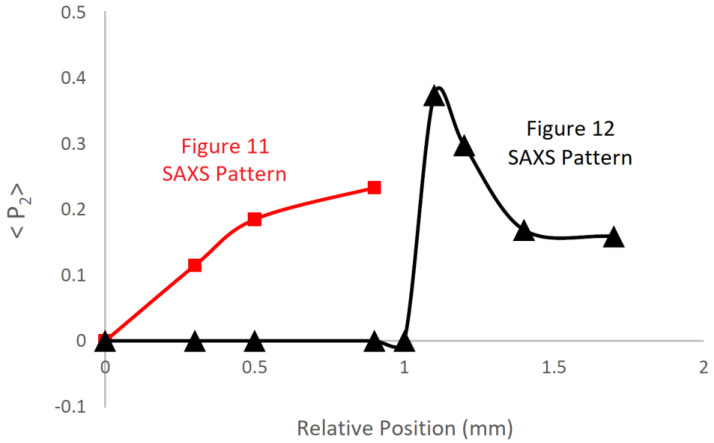
A plot of the level of preferred orientation of the chain-folded lamellar crystals in extruded strands of poly(ε-caprolactone) as a function of the position within the extrudate relative to the extrusion die exit. Here, ■ represents data evaluated from the SAXS patterns in Figure 11, ▲ represents data evaluated from the SAXS patterns in Figure 12.

**Figure 15 polymers-14-01638-f015:**
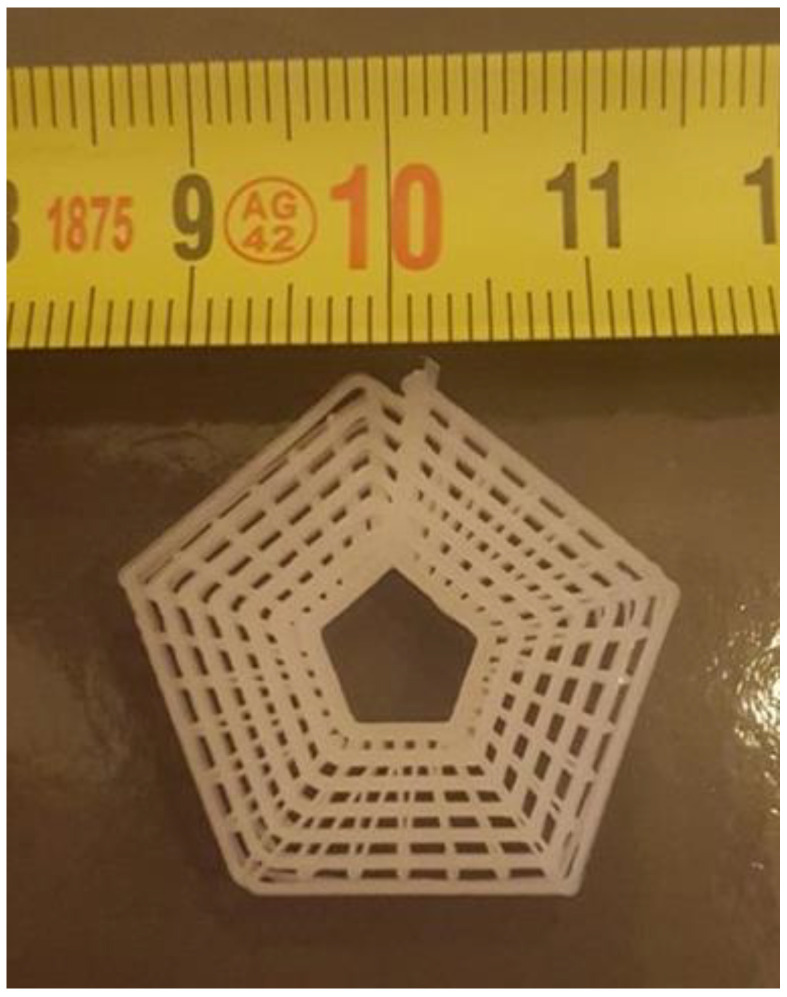
A photograph of a porous structure, typical of that used as a scaffold for tissue engineering, prepared by printing poly(ε-caprolactone) at a constant extrusion temperature with a deposition rate (collector velocity) which was proportional to the distance from the centre of the object, with all other print parameters fixed. As a consequence, the material on the outer part is stiffer than the central region. The height is about 2 cm.

## Data Availability

All raw data and the associated metadata were obtained as a result of public access experiments, and will be open access to all registered ALBA users after an initial embargo period of 3 years, during which time access is restricted to the experimental team (ALBA DATA POLICY). The experimental data identifiers are available from the corresponding author after the end of the embargo period.

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
