# Peer review of "Changing the Paradigm-Controlling Polymer Morphology during 3D Printing Defines Properties"

_polymers, 2022, doi:10.3390/polym14091638_

Round 1
Reviewer 1 Report
The work is novel in that the in situ examination technology and analysis are employed in the FDMed polymeric fabrication process.
However, the title and the keywords of the paper should be thoroughly changed. The current title reads like a review paper without specific designation towards the content of the research work. The keywords are not specific either and are thus meanningless.
Author Response
We thank the reviewers for their positive comments on the above manuscript and their suggestions for improving the ms. We have responded positively to each of the comments made and we list our replies and the modifications made to the manuscript.
Reviewer 1
However, the title and the keywords of the paper should be thoroughly changed. The current title reads like a review paper without specific designation towards the content of the research work. The keywords are not specific either and are thus meanningless.
Our belief is that as we wrote in the manuscript “Although the realisation that products could be produced from materials with suitable properties for application has been widely adopted in additive manufacturing, there has no real effort to further develop the technology to include patterns or texture of properties so as to be able to deliver the function of the product at the point of fabrication.” It is on that basis we have constructed the title, but we recognize that the comment of the reviewer , which is not reflected in the comments of reviewer 2, does have some merit and accordingly we have modified the title to be
“Changing the Paradigm – controlling polymer morphology during 3D printing defines properties”
which maintains the spirit of the original title whilst providing more information as requested by reviewer 1. Keywords also updated.
Reviewer 2 Report
In the current work, the group of G. R. Mitchell introduces and discusses the idea of having control over the texture/crystalline form of the polymeric materials obtained during fused position modelling (i.e. 3D printing). In this was particular shape/product made of a single polymer in addition to the shape may be able to have different physical properties cross its body as a result of different crystallisations taking place. This is indeed, quite progressive and thought provoking. The closest parallel one may draw is with metamaterials.
Although the idea is progressive, currently it is really a challenge for the authors or any other group to follow the crystallisation processes on a molecular level while manufacturing. However, to obtain as approximate an understanding as possible, the authors developed their own custom made 3d printer which they could integrate into the NCD-SWEET beamline (ALBA Synchrotron radiation source in Spain). It is then based on the SAXS patterns of the extrude that the authors notice that faster writing rates lead to higher degree of crystalline orientations, while for the lower writing rate the there is still crystallisation but it is largely isotropic. They provide a meaningful explanation for this which is that faster writing results in faster cooling for the extrude. The faster cooling provides little time for the formerly local liquid crystal directors to reorient and thus more ordered crystalline states of the extrude are obtained.
Overall the manuscript is well written and nicely illustrated. There are only a few minor typing errors due to formatting (e.g. “proper-ties” in line 35). Apart from that, the authors followed the classical article format for polymers. The classical format has a section for methods and then a separate one for results. In this work, the authors developed a custom made 3 d printer, which in my view is an engineering result and ideally should be acknowledged as such and be placed in the result section.
In addition, the authors placed a link for their data and results, however, the link leads to a pdf that discusses the beam time in ALBA. Any data that allows reproducibility of the work should be added. This also implies that the authors should provide sufficient information on how they have constructed their own 3d printer (i.e. specification of parts/components and assembly).
Once these issues are clarified, I think the overall manuscript is good for publication in Polymers.
Author Response
Reviewer 2
- A) We thank Reviewer 2 for their very positive comments on the ms. We agree that the development of a 3d printer able to fully function on the ALBA NCD-SWEET beamline is a major result of this work and so we have moved that component from the methods to the results and we have restructured the results section in to three sub-sections.
- B) We note the comments that Reviewer 2 made with respect to the data. The file the reviewer refers to is the Data Policy of ALBA and we have restructured the “Data Availability Statement” to make the meaning clearer and the availability of the data.
- C) In light of the comments regarding the 3D printer we have included high resolution CAD images of the 3d printer developed in this work in the supplementary materials.